# Sigma-1 Receptor (S1R) Interaction with Cholesterol: Mechanisms of S1R Activation and Its Role in Neurodegenerative Diseases

**DOI:** 10.3390/ijms22084082

**Published:** 2021-04-15

**Authors:** Vladimir Zhemkov, Michal Geva, Michael R. Hayden, Ilya Bezprozvanny

**Affiliations:** 1Department of Physiology, UT Southwestern Medical Center, Dallas, TX 75390, USA; Vladimir.Zhemkov@UTSouthwestern.edu; 2Prilenia Therapeutics Development LTD, Herzliya 4673304, Israel; Michal.Geva@prilenia.com (M.G.); Michael.Hayden@prilenia.com (M.R.H.); 3Centre for Molecular Medicine and Therapeutics, The University of British Columbia, Vancouver, BC V6H 3V5, Canada; 4Laboratory of Molecular Neurodegeneration, Peter the Great St Petersburg State Polytechnic University, 195251 St. Petersburg, Russia

**Keywords:** sigma-1 receptor, endoplasmic reticulum, mitochondria, contact sites, cholesterol, neurodegeneration, Huntington’s disease, Alzheimer’s disease, amyotrophic lateral sclerosis, drug target

## Abstract

The sigma-1 receptor (S1R) is a 223 amino acid-long transmembrane endoplasmic reticulum (ER) protein. The S1R modulates the activity of multiple effector proteins, but its signaling functions are poorly understood. S1R is associated with cholesterol, and in our recent studies we demonstrated that S1R association with cholesterol induces the formation of S1R clusters. We propose that these S1R-cholesterol interactions enable the formation of cholesterol-enriched microdomains in the ER membrane. We hypothesize that a number of secreted and signaling proteins are recruited and retained in these microdomains. This hypothesis is consistent with the results of an unbiased screen for S1R-interacting partners, which we performed using the engineered ascorbate peroxidase 2 (APEX2) technology. We further propose that S1R agonists enable the disassembly of these cholesterol-enriched microdomains and the release of accumulated proteins such as ion channels, signaling receptors, and trophic factors from the ER. This hypothesis may explain the pleotropic signaling functions of the S1R, consistent with previously observed effects of S1R agonists in various experimental systems.

## 1. Introduction

The sigma-1 receptor (S1R) is a 223 amino acid-long transmembrane protein residing in the endoplasmic reticulum (ER) [1,2,3]. S1R attracts significant attention as a potential drug target for treating neurological disorders [2,4,5,6] and cancers [6].

S1R is expressed at high levels in the CNS and specifically in the cortex, basal ganglia, and motor neurons of the spinal cord and brainstem [7,8,9,10]. The S1R is a chaperone protein that is enriched at the ER/mitochondria-associated membrane (MAM), where it plays an important role in the regulation of multiple cellular mechanisms and is key to maintaining neuronal function and health. This is further supported by human genetic studies, showing that complete loss of function (LOF) mutations in the S1R are associated with a juvenile form of amyotrophic lateral sclerosis/frontotemporal dementia (ALS/FTD), while partial LOF mutations cause late onset ALS [11,12,13,14]. Thus, there is a gene dosage relationship between S1R activity and the age of onset of ALS with the complete loss of S1R associated with the earliest age of onset. Additional LOF mutations in the S1R cause distal hereditary motor neuropathies (dHMN) [15,16,17,18,19]. Furthermore, S1R expression levels are reduced in sporadic ALS [20], Parkinson’s disease (PD), and Alzheimer’s disease (AD) patients [21,22]. In preclinical models, the genetic ablation of S1R in mice exacerbated pathology and phenotypic presentation of several neurological disorders [23,24,25]. These results suggest that S1R plays an important role in healthy neuronal physiology.

The first prototypic S1R agonist, SKF-10047, was identified in animal behavioral assays, which led to the proposed existence of sigma opioid receptors [26]. However, SKF-10047 binding to sigma binding sites was not blocked by naloxone, an opioid receptor antagonist, and displayed a different stereospecificity [27,28]. Subsequent cloning of sigma binding sites confirmed that they share no homology with opioid G protein-coupled receptors (GPCRs), as well as sharing little homology to any other mammalian protein [29,30]. Sequence analysis revealed homology with the fungi C7–C8 sterol isomerase. While the S1R does not have isomerase activity, it contains two sterol-like binding domains as part of its ligand-binding site [29]. Recent biochemical and structural analysis indicated that the S1R is a single transmembrane domain protein with a short cytoplasmic tail and a large luminal ligand-binding domain [31,32]. It is suggested that the S1R acts as a molecular chaperone, which can stabilize the native conformation of multiple client proteins in stress conditions [1,33,34]. The S1R can be activated with highly selective synthetic ligands with nanomolar affinity [34,35,36]. The identity of an endogenous ligand is under investigation, with endogenous steroids (pregnenolone, dehydroepiandrosterone sulfate (DHEA), progesterone) being the most likely candidate [37,38], and N,N-dimethyltryptamine [39], sphingolipids [40], and more recently, choline also investigated [41].

Despite its importance in physiology and disease, the biological function of S1R is poorly understood [3]. This protein is involved in many biological processes and signaling pathways including maintenance of calcium homeostasis [42,43,44,45], protein folding [42], stress-response [1,42,46,47], autophagy [48,49], and the regulation of cellular excitability [50,51,52]. The S1R modulates the activity of ion channels via protein–protein interaction [52,53]. The S1R mode of action is not coupled to any known signaling cascade and is more consistent with its role as a modulatory or adaptor protein, or, using a term first coined by Hayashi and Su, a “molecular chaperone” [34,42,54]. 

Several S1R-interacting partners have been identified and multiple recent reviews comprehensively summarized these S1R interactors and the S1R-induced modulation of their activities [2,3]. Apart from that, S1R is known to interact and mediate the clustering of cholesterol and ceramides in the ER, as shown in cell-based assays [55,56,57,58]. We recently demonstrated that S1R is associated with cholesterol-enriched clusters in the membranes using in vitro reconstitution approach [59]. 

In this review we propose a hypothesis that the biological functions of the S1R are mediated by its ability to form ER signaling cholesterol-enriched lipid microdomains, analogous to the lipid rafts in the plasma membrane [60]. 

## 2. Intracellular Localization of the S1R

S1R primarily resides in the ER membrane where it forms microdomains [42,59,61,62]. Its localization is in contrast to the uniform distribution pattern of ER markers, such as the Sec61b protein. A significant proportion of S1R is localized to MAMs, an ER sub-compartment closely associated with the mitochondria [42,59], in proximity to lipid droplets [63], and at the ER-plasma membrane (PM) junctions [59,64]. It is likely that S1R are localized to additional inter-organelle contact sites, but this has not been systematically investigated. 

MAMs are distinct from the rest of the ER as they contain enzymes involved in lipid synthesis, calcium signaling, cholesterol metabolism, and the ER stress-response pathways [65,66,67,68,69]. A detailed protein composition of MAMs was initially characterized by biochemical purifications [70,71] and more recently established using sophisticated proximity labeling approaches [72,73,74]. 

While the precise lipid composition of the MAMs has not yet been elucidated, recent evidence suggests that cholesterol and ceramide content is significantly higher in MAMs compared to the rest of the ER [57,75,76]. Therefore, MAMs can be thought of as specialized ER signaling domains characterized by unique protein and lipid compositions, similar to PM lipid rafts [60]. 

While PM lipid and protein heterogeneity was visualized using the giant plasma membrane-derived vesicle technique [77,78,79], only recently was a similar method to yield endomembrane-derived giant unilamellar vesicle (GUV)-like vesicles developed [80]. Using this approach, it was observed that certain, but not all ER contact sites (such as ER-mitochondria, ER-PM, and ER-lipid droplets) showed separation of the glycosylphosphatidylinositol (GPI) ER-targeted marker with a strong affinity for lipid-ordered phase [80]. Similar lipid and protein compartmentalization were recently observed at the inter-organelle contact sites in yeast [81], providing additional experimental evidence for microscopic lipid heterogeneity in the ER.

Binding immunoglobulin protein/glucose-regulated protein 78 (BiP/GRP78) was identified as a major S1R binding partner using pull-down experiments [42]. The S1R interacts with BiP in a calcium- and agonist-dependent manner. At high Ca^2+^ concentrations or in the absence of an agonist, the S1R forms a complex with BiP, keeping it in an inactive state. Therefore, BiP interaction contributes to S1R retention in the ER.

Under conditions of calcium depletion, or agonist activation, the S1R dissociates from BiP [42]. Agonist stimulation leads to redistribution of S1R from clusters to the ER, plasma membrane, and extracellular space [55,61]. In flotation assays, activation by an agonist causes the S1R to translocate from detergent-resistant to detergent-soluble fractions [55,61,63].

S1R activation releases the inhibitory interaction with BiP and allows for the S1R to interact with various partners inside and outside of the MAMs, including the inositol-1,4,5-triphosphate receptor type 3 (InsP_3_R3) [34,42]. Our recent results [59] and previous studies [57] suggest that a direct, high-affinity association of the S1R with cholesterol and ceramides may also contribute to S1R targeting by MAMs. 

## 3. Interaction of S1R with ER Membranes

S1R was shown to interact with cholesterol in vitro [55], suggesting that S1R association with cholesterol plays an important role in MAM targeting of the S1R [57] and in modulation of PM cholesterol levels [58]. 

Recently, we demonstrated that cholesterol promotes the formation of S1R domains in a model lipid bilayer system [59]. Using GUVs with reconstituted fluorescent-labeled S1R, we observed that cholesterol was sufficient to cause clustering of recombinant S1R in the absence of any other proteins [59]. This study further identified a novel cholesterol-binding site within the transmembrane (TM) domain of the S1R. Additionally, point mutations in the TM domain, which weaken the interaction of the S1R with cholesterol, result in the impaired redistribution of S1R into the entire ER network [59].

In vitro, S1R clustering was observed in a narrow range between 2.5% and 5.0% mol cholesterol [59], comparable with the cholesterol-dependence of sterol regulatory element binding protein-2: sterol regulatory element-binding protein cleavage-activating protein complex (SREBP-2:Scap) [82]. Our data suggests that S1R actively participates in the assembly of micrometer-size cholesterol-enriched microdomains. On the basis of these observations, we propose that S1Rs promote the formation and stabilization of MAM microdomains, and potentially other ER contact sites [59]. Consistent with this idea, the genetic deletion of S1R impairs MAM stability and results in a reduced number of contacts, as observed by electron microscopy (EM) and biochemical fractionation [43].

The S1R has an unusually long, single transmembrane domain [59]. Local “measurements” of bilayer thickness with transmembrane sensors showed that bilayers surrounding S1R domains are thicker [59]. In the crystal structure, the S1R molecules are organized as trimers, with their C-terminal ligand-binding domain partially embedded in the membrane [31,83]. These amphipathic helices at the C-terminus of the S1R are rich in aromatic residues and, thus, can play a role in additional stabilization of the bilayer structure, a phenomenon previously observed for other amphipathic helices [84,85]. 

Thus, we reasoned that the local membrane thickness is increased in S1R-formed ER microdomains [59], which likely plays a role in the sorting of ER membrane proteins to these domains [79,81].

## 4. The S1R Interactome

To understand the functional significance of S1R-formed microdomains in the ER, we performed an unbiased screen aimed at identifying proteins located in proximity to S1R in cells. Analysis of such a “S1R interactome” further clarifies the composition and biological function of these microdomains.

Our experiments utilized a Tet-inducible vector expressing S1R fused to the peroxidase APEX2 in a proximity labeling technique, which captures weak and transient interactions [86]. 

For that purpose, we generated a plasmid encoding the S1R and fused to a genetically engineered APEX2 peroxidase, under the Tet-On tetracycline-inducible promoter (S1R-APEX2). Proteins in close proximity of S1R-APEX2 were biotinylated (see Materials and Methods Section for details) and the biotinylated proteins were pulled down using streptavidin-agarose. The eluted proteins were separated by sodium dodecyl sulfate polyacrylamide gel electrophoresis (SDS-PAGE) and analyzed by mass spectrometry. Control non-induced cells were treated exactly the same way.

Using this approach, we identified several hundred proteins enriched in samples from S1R-APEX2-induced HeLa cells (Figure 1A, Appendix A). To analyze these hits, we first utilized the “cellular component” gene ontology (GO) term and determined that the majority of these proteins were ER membrane or luminal proteins (Table 1), consistent with the luminal localization of the S1R C-terminus [32,87]. Other enriched proteins were components of the extracellular matrix (ECM) or the plasma membrane (PM), and a small fraction were identified as Golgi and lysosomal proteins (Table 1). Since no S1R-APEX2 staining was detected on the plasma membrane by electron microscopy analysis ([32,87] and our unpublished observations), we hypothesized that the PM proteins and ECM components corresponded to a newly synthesized pool of proteins, that have not yet exited from the ER membrane compartment (full lists of ECM and cell surface proteins are provided in Appendix A, respectively).

Hits from the S1R-APEX2 screen were then analyzed on the basis of their “biological process” GO terms. The top GO terms identified in this analysis were protein folding, oxidation-reduction processes, extracellular matrix organization, response to ER stress, protein modification, and protein glycosylation (Table 2). 

More specific analysis was performed using a gene-set enrichment analysis (GSEA) [88] of these hits. GSE analysis (Figure 1B) revealed that the identified proteins (referred here by their UniProt names) were involved in: (1) the formation and shuffling of disulfide bonds (multiple protein disulfide isomerases protein disulfide isomerases (PDIs) PDIA3/4/5/6, thioredoxin-related transmembrane protein 3 (TMX3) and disulfide regenerating enzyme ERO1-like protein alpha (ERO1A)); (2) N- and O-linked glycosylation (including enzymes involved in multiple steps of the co-translational attachment of the dolichol-phosphate oligosaccharyl precursor (RPN1/2, STT3A/B), initial trimming of mannose chains (mannosyl-oligosaccharide glucosidase (MOGS); neutral alpha-glucosidase AB (GANAB)), and quality control and refolding of sugar chains (UDP-glucose:glycoprotein glucosyltransferase 1/2 (UGGG1/2) and GANAB); (3) the attachment of a glycosylphosphatidylinositol (GPI) anchor (PIGS, PIGO, GPAA1); and (4) in ER quality control including lectins, calnexin (CALX)/calreticulin (CALR), and machinery involved in targeting of misfolded proteins to the retrotranslocation complex (ER degradation-enhancing alpha-mannosidase-like proteins (EDEM2/3), ERLEC1, OS9 and protein SEL1 homolog (SEL1L)). No significant components of the ER ubiquitin-ligase complex were identified, except for SEL1L. 

## 5. The Functional Role of S1R Microdomains: A Hypothesis

The results obtained in our studies with a reconstituted S1R [59] and in the S1R-APEX2 screen (Figure 1) led us to propose a novel hypothesis regarding the biological function ofS1R in cells. We propose that S1R organizes cholesterol-enriched microdomains in the ER (Figure 2). We reason that these microdomains are analogous to lipid rafts in the plasma membrane [60] and that, similar to lipid rafts, these microdomains have unique lipid and protein compositions when compared to the rest of the ER membrane. We also reason that these microdomains are thicker than the rest of the ER [59]. Our data [59] suggest that these microdomains are preferentially formed at ER membrane contact sites such as MAMs. We propose that these microdomains serve as a platform for the post-translational modification (PTM) and maturation of proteins in the ER, on a pre-secretory stage. These platforms may also play a role in protein folding-modification processes that take place in the early stages of protein synthesis, before properly folded proteins can be sorted to post-ER organelles.

No major ER exit site markers were identified in the S1R-APEX2 screen (Figure 1A, Appendix A), suggesting they are localized outside of the S1R-formed microdomains. According to our hypothesis, PM proteins and proteins destined for secretion to the extracellular space are temporarily “trapped” in rigid and cholesterol-rich S1R-ER microdomains. While located in the S1R microdomains, these proteins can be processed through folding-modification cycles with the help of the enzymatic machinery residing in these domains. Strategic placement of S1R microdomains at MAMs ensures constant ATP supply for high-energy-dependent oxidative folding and protein modification reactions.

In agreement with this hypothesis, the experimental evidence indicates that membrane-bound calnexin and TMX are recruited to MAMs through palmitoylation, a known raft targeting mechanism [89,90]. Components of GPI machinery are localized to ER detergent-resistant membranes [91]. ER detergent-resistant membranes play an important role in assembly and secretion of viral particles [92,93]. The timeframe of protein maturation in the ER is known to be tightly controlled by mannose-trimming enzymes [94]. This timing can be especially important for transmembrane proteins (such as ion channels and receptors) that contain multiple transmembrane domains. Local thinning of the ER membrane was recently proposed to be required for the retrotranslocation of proteins though the ER-associated degradation (ERAD) mechanism [95,96]. Therefore, client proteins and enzymatic machinery localized to the “thick” S1R microdomains [59] are largely protected from a premature proteasomal degradation by ERAD.

## 6. The S1R as a Therapeutic Target for the Treatment of Neurodegenerative Diseases

The S1R is a well-established target for the treatment of neurodegenerative disorders, and it plays a key role in neurodegenerative diseases. Several S1R mutations have been identified to be associated with ALS and frontotemporal dementia (FTD) [97]. Two complete loss of function (LOF) mutations cause a juvenile form of ALS [12,43]. However, missense mutations that partially impair protein function are associated with an adult form of ALS, showing a dose response between the function of S1R and the age of onset of ALS [98]. Additional LOF mutations in S1R cause hereditary motor neuropathies [15,18]. Furthermore, some variants of the S1R gene are associated with increased risk for Alzheimer’s disease (AD) [99]. In addition, S1R expression levels are reduced in sporadic ALS [20], Parkinson’s disease (PD), and Alzheimer’s disease (AD) patients [21,100].

Further support for the role of S1R in neurodegenerative diseases comes from animal models. In preclinical models, genetic ablation of S1R (S1R−/−) in mice exacerbates pathology and phenotypic presentation of several neurological disorders. For example, S1R−/− mice display impairments in motor function and degeneration of motor neurons at 5 months of age [101]. AD mice (APP^sweInd^), which lack S1R, show enhanced behavioral and cognitive impairments, as well as a significant reduction in the levels of the brain-derived neurotrophic factor (BDNF) compared to APP^sweInd^ mice expressing the S1R [102]. In the ALS SOD1^G93A^ mouse model that also lacks S1R expression (SOD1^G93A^/S1R KO), disease progression is accelerated, as revealed by earlier weight loss and by a ~32% decrease in survival time relative to SOD1^G93A^ mice with normal S1R expression [101]. These results suggest that the S1R plays an important role in healthy neuronal physiology. 

S1R activation by agonists has demonstrated neuroprotective effects in multiple cellular and animal models of neurodegeneration (reviewed in [5,45,103,104]). Extensive evidence suggests that the mechanisms responsible for the neuroprotective effects of S1R agonists include the stabilization of Ca^2+^ signaling [42,44,105,106], an increase in the secretion of BDNF and the potentiation of BDNF-tropomyosin-related kinase B (TrkB) signaling [107,108,109,110,111,112], the stimulation of cyclic AMP-responsive element-binding protein (CREB)-mediated transcription [110], changes in the activity of the plasma membrane ion channels and receptors [113,114,115,116], the potentiation of the N-Methyl-D-aspartate (NMDA) receptor response [113,117,118,119], and an improvement in mitochondrial function [43,120,121,122]. 

For example, we and others have shown that the highly selective and potent S1R agonist pridopidine restores the dysregulated ER Ca^2+^ signaling and enhances spine density in Huntington’s disease (HD) and Alzheimer’s disease (AD) neurons [44,45,123]. Furthermore, S1R activation by pridopidine enhances synaptic plasticity in HD cortical neurons [123] and exhibits a robust neuroprotective effect against mutant huntingtin-induced cell death in mice’s primary HD neurons and in HD patient-derived induced-pluripotent stem cells (iPSCs) [124]. Pridopidine has been found to upregulate BDNF secretion, potentiate BDNF-TrkB signaling, and enhance BDNF axonal transport in several different models of neurodegenerative diseases including HD and ALS [109,110,125]. Pridopidine has shown protective effects on several mitochondrial functions in various human and mouse models of HD. In primary HD neurons, pridopidine enhances mitochondria-ER tethering and restores mitochondrial function as measured by increased ATP production, respiration, and mitochondrial membrane potential [126]. All these effects are exquisitely mediated by the activation of the S1R, as either a genetic deletion of the S1R or a pharmacological inhibition using an S1R antagonist, completely abolishes pridopidine’s neuroprotective effects, as shown in the studies mentioned above [126].

Recent clinical studies have shown the potential efficacy of the selective S1R agonist pridopidine in HD patients, demonstrating maintenance or slowing the decline of the patient’s functional capacity [127,128]. The non-selective S1R/Muscarinic (M1R) agonist blarcamesine shows a potential beneficial effect in AD [129]. Clinical pivotal studies with pridopidine are currently ongoing for HD and ALS (NCT04556656, NCT04297683). Blarcamesine is currently being evaluated for AD, Rett syndrome, and PD dementia patients (NCT04314934, NCT04304482, NCT04575259). Results of completed clinical trials of S1R agonists in variety of disorders have been comprehensively summarized in recent reviews [4,130]. 

How can the activation of S1R exert such pleotropic and variable effects on cellular signaling? We propose that the agonist activation of S1R results in the remodeling of S1R microdomains (Figure 2). In our experiments [59] and in published studies [55,61], the activation of S1R has resulted in the disassembly of the S1R oligomers. We propose that agonists cause partial disassembly and remodeling of S1R microdomains in the ER, leading to a rapid release of mature proteins that are trapped in these microdomains.

Consistent with this hypothesis, increases in the levels of PM proteins are often observed following S1R stimulation with an agonist. For example, the S1R agonist SKF-10047 increased the plasma membrane fraction of the GluN1, GluN2A, and GluN2b NMDAR subunits [118]. Cocaine increased the PM fraction of Kv1.2 [50,51]. Cell surface expression of programmed cell death 1 ligand 1 (PD-L1) was increased after the agonist stimulation, and lower levels of PD-L1 were observed in S1R knockdown (KD) cells [131]. On the other hand, S1R knockdown (KD) decreased the stability and levels of the mature human ether-à-go-go-related gene (hERG), as well as hERG currents [132]. A shorter protein half-life was observed for InsP_3_R3 in MAMs in S1R KD cells [42], and the turnover of p35, which is dependent on protein myrostyilation, was lower in S1R deleted cells [133]. In contrast, however, the stability of UDP-galactose:ceramide galactosyltransferase (UGT8) was increased in S1R deleted cells [134].

The same idea applies not only to PM proteins, but also to secreted proteins. BDNF release increased after treatment with the S1R agonists pridopidine and cutamesine in B104 cells and in astrocyte cultures [111,135]. Levels of secreted BDNF and glial cell-derived neurotrophic factor (GDNF) were increased in substantia nigra after pridopidine and sigma receptor agonist PRE-084 treatment in the experimental Parkinsonism model [112,136]. Moreover, it was shown that BDNF processing is modulated by S1R at the post-translational stage [135], in line with our hypothesis. 

On the basis of these results, we propose that there is a “reserve pool” of PM and secreted proteins which accumulate in cholesterol-rich ER microdomains. S1R agonists facilitate the remodeling of these microdomains and the rapid release of PM and secreted proteins, resulting in a robust response to stress and in neuroprotective effects. 

Additional neuroprotective effects of S1R activation may also be related to the modulation of ER Ca^2+^ signaling, in particular inositol trisphosphate receptor (InsP_3_R)-mediated Ca^2+^ signaling [42,44,137] and ER-mitochondrial Ca^2+^ transfer [42,43]. The possible explanations for the observed potentiation of InsP_3_-induced Ca^2+^ release [137,138,139] can include the direct effects of cholesterol on the activity of the InsP_3_Rs, the modulation of ER Ca^2+^ content through cholesterol regulation of sacro/endoplasmic reticulum Ca^2+^ (SERCA) pump [140,141,142], the removal of the ankyrin inhibition of InsP_3_Rs [137], and the redistribution of InsP_3_Rs from MAMs to the peripheral ER [42,43]. 

This proposed model may explain the pleiotropic effects of S1R agonists and provide appropriate context for the development of S1R-targeting therapeutic agents.

## 7. Materials and Methods

### 7.1. Construct Design and Molecular Cloning

For cloning the S1R-APEX2 fusion gene, APEX2 (https://www.addgene.org/92158, accessed on 1 March 2019) and human S1R (NM_005866) genes were amplified by PCR using the following primers: S1R-EcoRI-F 5′ TAAATGAATTCATGCAGTGGGCCGTGGGCCGG, S1R-NotI-R 5′ GATGCGGCCGCAGGGTCCTGGCCAAAGAGGTAGGT, APEX2-NotI-F 5′ ATCGCGGCCGCCACCATGGACTACAAG, APEX2-BamHI-Rev 5′ ATTGGATCCTTAGGCATCAGCAAACCCAAGCTC. The NotI site was introduced to the APEX2 5′ primer and to the S1R 3′ primer. PCR products were ligated using T4 ligase (NEB, Ipswich, MA, USA) and amplified using the outer primers S1R-EcoRI-F and APEX2-BamHI-Rev to produce the fusion gene S1R-APEX2. The resulting S1R-APEX2 gene was cloned into pTRE-3g expression vector (Takara, Kusatsu, Japan) using EcoRI and BamHI cloning sites.

### 7.2. Generation of Stable HeLa Cell Lines

For the generation of stable lines, pTRE-3g-S1R-APEX2 plasmid was transfected in HeLa Tet-On inducible cell line (kindly provided by Phillip Thomas’s lab, UTSW) together with a linear puromycin-resistant marker (Takara, Kusatsu, Japan). Stable doxycycline-responsive monoclonal lines were isolated and expanded. The induction of the S1R-APEX2 fusion protein was confirmed by Western blot analysis using anti-APEX2 horseradish peroxidase-conjugate (ab192968, 1:1000, Abcam, Cambridge, UK) and anti-S1R (B-5, 1:300, Santa-Cruz, CA, USA) antibodies.

### 7.3. Proximity Biotinylation Experiments 

For the APEX2-based proximity-labeling experiments, we followed a procedure described in [143]. Briefly, for each experiment, the S1R-APEX2 HeLa cells were cultured on six 10 cm^2^ dishes. S1R-APEX2 production was induced by the addition of 1 μM doxycycline to the medium. After 72 h post-induction, the cells were incubated in 500 μM biotin-phenol (Iris Biotech, Marktredwitz, Germany) in complete medium at 37 °C for 1 h. Then, proteins were labeled by the addition of 1 mM H_2_O_2_ for 1 min and quenched with 10 mM sodium ascorbate, 5 mM Trolox, and 10 mM sodium azide in phosphate buffered saline (PBS). Cells were lysed in radioimmunoprecipitation assay buffer (RIPA) buffer (50 mM Tris-HCl pH = 7.4, 150 mM NaCl, 0.1% SDS, 0.5% sodium deoxycholate, 1% Triton X-100, 1× complete protease inhibitor cocktail) for 15 min at 4 °C on a rocker shaker. After centrifugation at 14,000× *g* for 10 min, 1 mL of lysate was mixed with 50 μL of streptavidin-agarose (Pierce) and incubated at 4 °C for 4 h on a rotary shaker. Resin was washed twice with 1 mL of RIPA buffer, once with 1 M KCl, once with 0.1 M Na_2_CO_3_, once with 2 M urea in 25 mM Tris-HCl pH = 8.0, and twice with RIPA buffer. Proteins were eluted by boiling beads in 50 μL of 2× SDS Laemmli loading buffer plus 2 mM biotin. Protein biotinylation was confirmed by Western blot analysis using streptavidin-HRP (7403, 1:20,000, Abcam, Cambridge, UK) antibody. Experiments were performed in duplicates with non-induced cells serving as the control.

### 7.4. Protein Identification by LC-Tandem Mass Spectrometry and Data Analysis

Eluates were loaded on a 12% gel (BioRad, Hercules, CA, USA). Gels were stained with Coomassie blue. Stained 1-cm bands were cut out of gels, sliced into 1-mm cubes and transferred to 1.5 mL Eppendorf tubes for submission. Raw mass spectrometry data was pre-proceeded and provided by UT Southwestern Proteomics Core. The statistically significant protein hits fold enrichment (calculated by diving a sum of spectral index values in doxycycline-induced samples vs. control samples) and *p-*values were calculated for each identified protein with at least three peptide sequences. Hits were selected by applying the following criteria: fold change >2.5 and *p-*value < 0.05. For gene ontology analyses, UniProt IDs were converted to gene names and analyzed by DAVID bioinformatic resources v6.8 (Laboratory of Human Retrovirology and Immunoinformatics, Frederick National Laboratory) [144]. Gene-set enrichment analyses (GSEA) were performed using GSEA v.4.1.0 (UC San Diego/Broad Institute) [88] software using the indicated molecular signature databases according to the instructions (https://www.gsea-msigdb.org/gsea, accessed on 1 March 2020). Data was visualized in R using the ROTS [145] package and GSEA software [88]. 

## Figures and Tables

**Figure 1 ijms-22-04082-f001:**
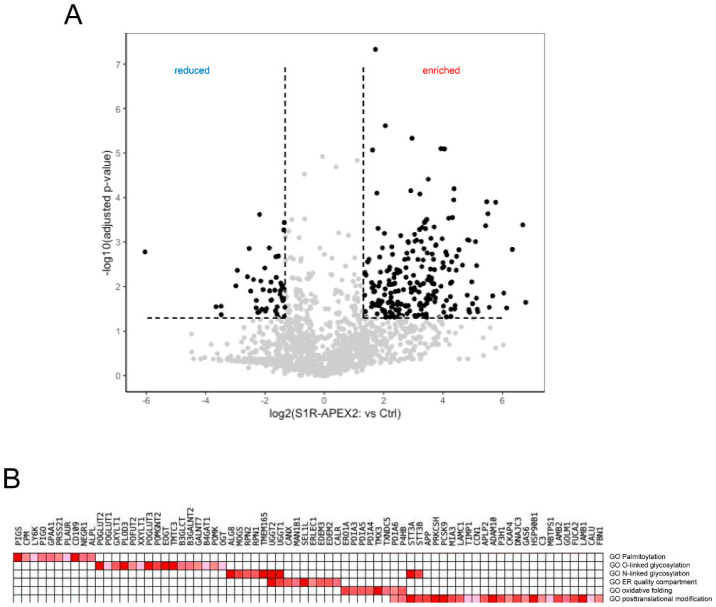
Identification of sigma-1 receptor (S1R) interactome. (**A**) The volcano plot of proteins identified in the S1R-APEX2 proteomic screen: x-axis—log2 fold change in S1R-APEX2-expressing HeLa cell line vs. non-induced control cells; y-axis—log10 adjusted *p-*value; (**B**) Gene-set enrichment analysis (GSEA) identified S1R clusters as sites for protein folding and post-translational modifications; the hits identified in the screen were compared to manually curated lists using GSEA.

**Figure 2 ijms-22-04082-f002:**
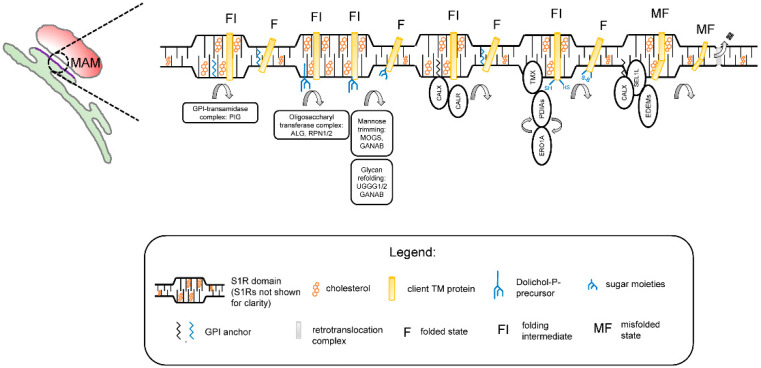
Proposed function of S1R microdomains in the endoplasmic reticulum (ER). S1R microdomains are formed in the ER membranes and enriched in MAMs. S1R microdomains are characterized by higher cholesterol content and thicker bilayer structure when compared to the rest of the ER membrane. In S1R domains, protein folding-modification machinery is compartmentalized and allows a protein to proceed from a folding intermediate (FI) state to a fully folded (F) state. Client proteins (shown here as a single transmembrane helix for simplicity), which can include ER resident proteins, components of extracellular matrix, and PM channels/receptors, are recruited to these clusters. Folding machinery include enzymes involved in the following processes: GPI transamidase complex (PIG) catalyzing the transfer of a GPI moiety to nascent protein chains; initial glycosylation steps (components of oligosacharyl transferase complex alpha-1,3/1,6-mannosyltransferase (ALG), recognition particle 1/2 (RPN1/2)), initial mannose trimming steps (MOGS, GANAB), and glycan refolding (UGGG1/2, GANAB); ER lectins involved in protein quality control and refolding (calnexin (CALX)/calreticulin (CALR)); oxidative folding enzymes (TMX and protein disulfide isomerases); and finally, quality control lectins guiding unfolded proteins for recognition by the ER-associated protein degradation pathway (ERAD). Agonists affect the S1R oligomerization state and result in partial disassembly of S1R domains, leading to release of proteins from S1R clusters.

**Table 1 ijms-22-04082-t001:** Cellular component gene oncology (GO) classification of top hits from S1R-APEX2 screen.

GO_TERM Cellular Component	% Fraction in Screen (Number of Hits)	*p*-Value
Total hits, fold change > 2.5, *p*-value < 0.05	100 (219)	-
ER membrane	25.2 (55)	1.5 × 10^−24^
ER lumen	20.2 (44)	8.1 × 10^−43^
Extracellular space	18.3 (40)	1.9 × 10^−7^
Cell surface	13.8 (30)	1.3 × 10^−11^
Golgi	8.7 (19)	1.5 × 10^−2^
Lysosome	6.0 (13)	3.7 × 10^−10^
Nuclear envelope	4.1 (9)	6.5 × 10^−4^
ER-Golgi intermediate compartment	4.2 (7)	1.6 × 10^−4^
ER quality compartment	2.8 (6)	2.7 × 10^−7^

**Table 2 ijms-22-04082-t002:** Biological process GO classification of top hits from S1R-APEX2 screen.

GO-SLIM_TERM Biological Process	Number of Hits	*p*-Value
Protein folding in the ER	10	5.4 × 10^−8^
Extracellular matrix organization	7	5.4 × 10^−8^
ERAD pathway	5	2.8 × 10^−4^
ER stress-response	5	1.9 × 10^−5^
N-linked glycosylation	4	4.8 × 10^−6^
O-linked glycosylation	3	8.8 × 10^−4^
ER quality control	3	3.5 × 10^−4^

## Data Availability

Results of S1R-APEX2 screen are available as Appendix A.

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
