# Peer review of "Sigma-1 Receptor (S1R) Interaction with Cholesterol: Mechanisms of S1R Activation and Its Role in Neurodegenerative Diseases"

_ijms, 2021, doi:10.3390/ijms22084082_

Round 1

Reviewer 1 Report

Zhemkov et al. summarized a finding  that the S1R interacts with cholesterol and suggest a working hypothesis in this review. Though the authors wrote the article as a review, they present some novel data. Thus this article might goe well as a brief communication.

The working hypothesis is interesting and brings a novel perspective to the molecular mechanisms of the chaperone S1R. However, few minor comments could improve the manuscript:

  1. The authors did not define all abbreviations in the manuscript such as:  BiP/GRP78 (line 106) and IP3R3 (line 116).
  2. It would be nice to introduce endogenous and exogenous ligands of the S1R.
  3. Though in the title the authors make it clear that they focus on neurodegenerative diseases, my question is here why is that? It seems to me that the suggested S1R mechanism is more general and disfunction of the receptor may also affect other tissues. Is the S1R also correlated to other diseases?
  4. It would be nice to have a table that summarizes the clinical trials.
  5. If I understood correctly, then different ligands release different proteins. Have the authors any explanation for this?
  6. It would be good if the authors would describe more in detail their hypothesis how S1R affects the IP3R-mediated Ca2+ signaling.
  7. In M&M, the following points would be good to include primer sequences.

Author Response

Zhemkov et al. summarized a finding that the S1R interacts with cholesterol and suggest a working hypothesis in this review.

Though the authors wrote the article as a review, they present some novel data. Thus, this article might go well as a brief communication.

Author: We thank the reviewer and agree that this article is better described as a brief communication.

The working hypothesis is interesting and brings a novel perspective to the molecular mechanisms of the chaperone S1R. However, few minor comments could improve the manuscript:

  1. The authors did not define all abbreviations in the manuscript such as:  BiP/GRP78 (line 106) and IP3R3 (line 116).

Author: We agree and according to the reviewer’s suggestion we have now defined abbreviations and added the full protein names.

  1. It would be nice to introduce endogenous and exogenous ligands of the S1R.

Author: Thank you for this comment. We have now added a paragraph discussing synthetic and potential endogenous S1R ligands in the introduction lines 61-65.

  1. Though in the title the authors make it clear that they focus on neurodegenerative diseases, my question is here why is that?

Author:

We thank the reviewer for this important comment and the opportunity to clarify.   We added the following discussion in the section 6 of the masnuscript (lines 317-337):

The S1R plays a key role in neurodegenerative diseases. Evidence for this can be learned from genetic studies. Several S1R mutations have been identified to be associated with ALS and frontotemporal dementia (FTD)  [1]. Two complete Loss Of Function (LOF) mutations in the highly conserved transmembrane domain cause a juvenile form of ALS [2, 3]. However, missense mutations that partially impair protein function are associated with an adult form of ALS, showing a dose response between function of the S1R and age of onset of ALS[4]. Additional LOF mutations in S1R cause hereditary motor neuropathies [5, 6]. Furthermore, some variants of the S1R gene are associated with increased risk for Alzheimer’s disease (AD) [7].  In addition, S1R expression levels are reduced in sporadic ALS [8], Parkinson’s disease (PD) and Alzheimer’s disease (AD) patients [9, 10].

Further support for the role of S1R in neurodegenerative diseases comes from animal models. In preclinical models, genetic ablation of the S1R (S1R-/-) in mice exacerbates pathology and phenotypic presentation of several neurological disorders. For example, S1R -/- mice display impairments in motor function and degeneration of motor neurons at 5 months of age [11]. In AD mice (APPsweInd), which also lack S1R show enhanced behavioral and cognitive impairments as well as a significant reduction in the levels of the Brain Derived Neurotrophic Factor (BDNF) compared to APPsweInd  mice expressing the S1R [12].  In the ALS SOD1G93A mouse model that also lack S1R expression (SOD1G93A/S1R KO), disease progression is accelerated, as revealed by earlier weight loss and by ~32% decrease in survival time relative to SOD1G93A mice with normal S1R expression [11]. These results suggest that the S1R plays an important role in healthy neuronal physiology.

It seems to me that the suggested S1R mechanism is more general and disfunction of the receptor may also affect other tissues.

Is the S1R also correlated to other diseases?

We agree with the reviewer, that the S1R can be of relevance to other therapeutic areas. For example, the S1R may be upregulated in cancer cells. Cancer therapy studies are currently assessing the effects of S1R antagonists, and not S1R agonists.  As this subject utside of the main scope of the manuscript, we refer reader to recent excellent review on this subject on line 33:  (Kim, F. J.; Maher, C. M., Sigma1 Pharmacology in the Context of Cancer. Handb Exp Pharmacol 2017, 244, 237-308).  

  1. It would be nice to have a table that summarizes the clinical trials.

Author:

We thank the reviewer for this interesting suggestion.  We agree with the reviewer that this is valuable information and have therefore referred to a few recent reviews which comprehensively cover clinical trials of S1R ligands [13, 14].

  1. If I understood correctly, then different ligands release different proteins. Have the authors any explanation for this?

Author: 

Thank you for raising this important question. Natural ligands for the S1R are not yet elucidated,. Various groups have studies different prototypic S1R agonists, different cell types and focused on different signaling proteins [15-25].

We hope that the hypothesis we present in our publication will stimulate systematic investigation into this question.  Currently there are no data to support a conclusion to this question.  

  1. It would be good if the authors would describe more in detail their hypothesis how S1R affects the IP3R-mediated Ca2+ signaling.

Author: 

Thank you for this important suggestion. We included more detailed discussion on the different potential ways by which the S1R regulates IP3R Ca2+ signaling (lines 411-416).

  1. In M&M, the following points would be good to include primer sequences.

Author:  Thank you for this comment. We have added the sequences of the cloning primers to the M&M section.

References :

  1. Luty, A. A.; Kwok, J. B.; Dobson-Stone, C.; Loy, C. T.; Coupland, K. G.; Karlstrom, H.; Sobow, T.; Tchorzewska, J.; Maruszak, A.; Barcikowska, M.; Panegyres, P. K.; Zekanowski, C.; Brooks, W. S.; Williams, K. L.; Blair, I. P.; Mather, K. A.; Sachdev, P. S.; Halliday, G. M.; Schofield, P. R., Sigma nonopioid intracellular receptor 1 mutations cause frontotemporal lobar degeneration-motor neuron disease. Ann Neurol 2010, 68, (5), 639-49.
  2. Watanabe, S.; Ilieva, H.; Tamada, H.; Nomura, H.; Komine, O.; Endo, F.; Jin, S.; Mancias, P.; Kiyama, H.; Yamanaka, K., Mitochondria-associated membrane collapse is a common pathomechanism in SIGMAR1- and SOD1-linked ALS. EMBO Mol Med 2016, 8, (12), 1421-1437.
  3. Al-Saif, A.; Al-Mohanna, F.; Bohlega, S., A mutation in sigma-1 receptor causes juvenile amyotrophic lateral sclerosis. Ann Neurol 2011, 70, (6), 913-9.
  4. Izumi, Y.; Morino, H.; Miyamoto, R.; Matsuda, Y.; Ohsawa, R.; Kurashige, T.; Shimatani, Y.; Kaji, R.; Kawakami, H., Compound heterozygote mutations in the SIGMAR1 gene in an oldest-old patient with amyotrophic lateral sclerosis. Geriatr Gerontol Int 2018, 18, (10), 1519-1520.
  5. Gregianin, E.; Pallafacchina, G.; Zanin, S.; Crippa, V.; Rusmini, P.; Poletti, A.; Fang, M.; Li, Z.; Diano, L.; Petrucci, A.; Lispi, L.; Cavallaro, T.; Fabrizi, G. M.; Muglia, M.; Boaretto, F.; Vettori, A.; Rizzuto, R.; Mostacciuolo, M. L.; Vazza, G., Loss-of-function mutations in the SIGMAR1 gene cause distal hereditary motor neuropathy by impairing ER-mitochondria tethering and Ca2+ signalling. Hum Mol Genet 2016, 25, (17), 3741-3753.
  6. Li, X.; Hu, Z.; Liu, L.; Xie, Y.; Zhan, Y.; Zi, X.; Wang, J.; Wu, L.; Xia, K.; Tang, B.; Zhang, R., A SIGMAR1 splice-site mutation causes distal hereditary motor neuropathy. Neurology 2015, 84, (24), 2430-7.
  7. Uchida, N.; Ujike, H.; Tanaka, Y.; Sakai, A.; Yamamoto, M.; Fujisawa, Y.; Kanzaki, A.; Kuroda, S., A variant of the sigma receptor type-1 gene is a protective factor for Alzheimer disease. Am J Geriatr Psychiatry 2005, 13, (12), 1062-6.
  8. Prause, J.; Goswami, A.; Katona, I.; Roos, A.; Schnizler, M.; Bushuven, E.; Dreier, A.; Buchkremer, S.; Johann, S.; Beyer, C.; Deschauer, M.; Troost, D.; Weis, J., Altered localization, abnormal modification and loss of function of Sigma receptor-1 in amyotrophic lateral sclerosis. Hum Mol Genet 2013, 22, (8), 1581-600.
  9. Jansen, K. L.; Faull, R. L.; Storey, P.; Leslie, R. A., Loss of sigma binding sites in the CA1 area of the anterior hippocampus in Alzheimer's disease correlates with CA1 pyramidal cell loss. Brain Res 1993, 623, (2), 299-302.
  10. Mishina, M.; Ishiwata, K.; Ishii, K.; Kitamura, S.; Kimura, Y.; Kawamura, K.; Oda, K.; Sasaki, T.; Sakayori, O.; Hamamoto, M.; Kobayashi, S.; Katayama, Y., Function of sigma1 receptors in Parkinson's disease. Acta Neurol Scand 2005, 112, (2), 103-7.
  11. Mavlyutov, T. A.; Epstein, M. L.; Andersen, K. A.; Ziskind-Conhaim, L.; Ruoho, A. E., The sigma-1 receptor is enriched in postsynaptic sites of C-terminals in mouse motoneurons. An anatomical and behavioral study. Neuroscience 2010, 167, (2), 247-55.
  12. Maurice, T.; Strehaiano, M.; Duhr, F.; Chevallier, N., Amyloid toxicity is enhanced after pharmacological or genetic invalidation of the sigma1 receptor. Behav Brain Res 2018, 339, 1-10.
  13. Ye, N.; Qin, W.; Tian, S.; Xu, Q.; Wold, E. A.; Zhou, J.; Zhen, X. C., Small Molecules Selectively Targeting Sigma-1 Receptor for the Treatment of Neurological Diseases. J Med Chem 2020, 63, (24), 15187-15217.
  14. Maurice, T., Bi-phasic dose response in the preclinical and clinical developments of sigma-1 receptor ligands for the treatment of neurodegenerative disorders. Expert Opin Drug Discov 2020, 1-17.
  15. Pabba, M.; Wong, A. Y.; Ahlskog, N.; Hristova, E.; Biscaro, D.; Nassrallah, W.; Ngsee, J. K.; Snyder, M.; Beique, J. C.; Bergeron, R., NMDA receptors are upregulated and trafficked to the plasma membrane after sigma-1 receptor activation in the rat hippocampus. J Neurosci 2014, 34, (34), 11325-38.
  16. Kourrich, S.; Hayashi, T.; Chuang, J. Y.; Tsai, S. Y.; Su, T. P.; Bonci, A., Dynamic interaction between sigma-1 receptor and Kv1.2 shapes neuronal and behavioral responses to cocaine. Cell 2013, 152, (1-2), 236-47.
  17. Delint-Ramirez, I.; Garcia-Oscos, F.; Segev, A.; Kourrich, S., Cocaine engages a non-canonical, dopamine-independent, mechanism that controls neuronal excitability in the nucleus accumbens. Mol Psychiatry 2020, 25, (3), 680-691.
  18. Maher, C. M.; Thomas, J. D.; Haas, D. A.; Longen, C. G.; Oyer, H. M.; Tong, J. Y.; Kim, F. J., Small-Molecule Sigma1 Modulator Induces Autophagic Degradation of PD-L1. Mol Cancer Res 2018, 16, (2), 243-255.
  19. Crottes, D.; Martial, S.; Rapetti-Mauss, R.; Pisani, D. F.; Loriol, C.; Pellissier, B.; Martin, P.; Chevet, E.; Borgese, F.; Soriani, O., Sig1R protein regulates hERG channel expression through a post-translational mechanism in leukemic cells. J Biol Chem 2011, 286, (32), 27947-58.
  20. Hayashi, T.; Su, T. P., Sigma-1 receptor chaperones at the ER-mitochondrion interface regulate Ca(2+) signaling and cell survival. Cell 2007, 131, (3), 596-610.
  21. Tsai, S. Y.; Pokrass, M. J.; Klauer, N. R.; Nohara, H.; Su, T. P., Sigma-1 receptor regulates Tau phosphorylation and axon extension by shaping p35 turnover via myristic acid. Proc Natl Acad Sci U S A 2015, 112, (21), 6742-7.
  22. Fujimoto, M.; Hayashi, T.; Urfer, R.; Mita, S.; Su, T. P., Sigma-1 receptor chaperones regulate the secretion of brain-derived neurotrophic factor. Synapse 2012, 66, (7), 630-9.
  23. Malik, M.; Rangel-Barajas, C.; Sumien, N.; Su, C.; Singh, M.; Chen, Z.; Huang, R. Q.; Meunier, J.; Maurice, T.; Mach, R. H.; Luedtke, R. R., The effects of sigma (sigma1) receptor-selective ligands on muscarinic receptor antagonist-induced cognitive deficits in mice. Br J Pharmacol 2015, 172, (10), 2519-31.
  24. Francardo, V.; Geva, M.; Bez, F.; Denis, Q.; Steiner, L.; Hayden, M. R.; Cenci, M. A., Pridopidine Induces Functional Neurorestoration Via the Sigma-1 Receptor in a Mouse Model of Parkinson's Disease. Neurotherapeutics 2019, 16, (2), 465-479.
  25. Francardo, V.; Bez, F.; Wieloch, T.; Nissbrandt, H.; Ruscher, K.; Cenci, M. A., Pharmacological stimulation of sigma-1 receptors has neurorestorative effects in experimental parkinsonism. Brain 2014, 137, (Pt 7), 1998-2014.

Reviewer 2 Report

IJMS -1178267 Review

Sigma-1 receptor (S1R) interaction with cholesterol: mechanisms of S1R activation and its role in neurodegenerative diseases.

Authors: Vladimir Zhemkov, Michal Geva,, Michael R. Hayden, and Ilya Bezprozvanny

How original is the topic? What does it add to the subject area compared with other published material?

Despite numerous Neuronal Sigma-1 Receptors based reviews in literature and by this group (Front. Neurosci., 28 August 2019 | https://doi.org/10.3389/fnins.2019.00862) the specific information of S1R and its molecular and structural details were written in this article and advances the field of neurodegeneration with the information presented. Some of the highlights of this review are :

  1. S1R association with cholesterol induces the formation of S1R clusters
  2. S1R agonists enable the disassembly of these cholesterol-enriched microdomains and the release of accumulated proteins such as ion channels, signaling receptors, and trophic factors from the ER.
  3. genetic ablation of the S1R in mice exacerbates pathology and phenotypic presentation of several neurological disorders
  4. Despite its importance in physiology and disease, the biological function of S1R is poorly understood
  5. The S1R modulates the activity of ion channels via protein-protein interaction
  6. The S1R mode of action is not coupled to any known signaling cascade and is more consistent with its role as a modulatory or adaptor protein, or, in a term first coined by Hayashi and Su, a “molecular chaperone”

Based on these points it is important to give a overview of the research adavances in to this important biological target and hence this review is very essential.

  1. Functional Role Of S1R Microdomains: A Hypothesis
  2. The S1r As A Therapeutic Target For The Treatment Of Neurodegenerative Diseases

Are very important for further studies in this area.

Is the paper well written? Is the text straightforward and easy to read?

  1. Paper well written
  2. Text if flowing and easy to read

Is there some point/hypothesis/idea in the manuscript that you do not agree with?

Is this a review or research article if this is a review why M&M if there is a new data provided then it should be classified properly and reformatted for the review or research.

Are the conclusions consistent with the evidence and arguments presented? Do they address the main question posed?

Interesting results :

Depends on reformatting

Minor comments (Specific comments referring to line numbers, tables or figures.)

None might come up in revisions

Author Response

Despite numerous Neuronal Sigma-1 Receptors based reviews in literature and by this group (Front. Neurosci., 28 August 2019 | https://doi.org/10.3389/fnins.2019.00862) the specific information of S1R and its molecular and structural details were written in this article and advances the field of neurodegeneration with the information presented.

Some of the highlights of this review are:

  1. S1R association with cholesterol induces the formation of S1R clusters
  2. S1R agonists enable the disassembly of these cholesterol-enriched microdomains and the release of accumulated proteins such as ion channels, signaling receptors, and trophic factors from the ER.
  3. Genetic ablation of the S1R in mice exacerbates pathology and phenotypic presentation of several neurological disorders
  4. Despite its importance in physiology and disease, the biological function of S1R is poorly understood
  5. The S1R modulates the activity of ion channels via protein-protein interaction
  6. The S1R mode of action is not coupled to any known signaling cascade and is more consistent with its role as a modulatory or adaptor protein, or, in a term first coined by Hayashi and Su, a “molecular chaperone”

Based on these points it is important to give an overview of the research advances into this important biological target and hence this review is very essential.

  1. Functional Role Of S1R Microdomains: A Hypothesis
  2. The S1r As A Therapeutic Target For The Treatment Of Neurodegenerative Diseases

Are very important for further studies in this area.

Author:  We thank the reviewer for these kind words and the recognition on the importance of our work.

Ref 2, comment 2: Is this a review or research article if this is a review why M&M if there is a new data provided then it should be classified properly and reformatted for the review or research.

Author:

: We thank reviewer for the positive evaluation of the paper. We agree with this comment and have changed the article type to “Short communication”.

Round 2

Reviewer 2 Report

As this is revision and the authors agreed to Short communication I recommend the paper to be accepted in present form.